# Reward Translation via Reward Machine in Semi-Alignable MDPs

**Yun Hua** [* 1]  **Haosheng Chen** [* 2]  **Wenhao Li** [3]  **Bo Jin** [3]  **Baoxiang Wang** [4]  **Hongyuan Zha** [4]  **Xiangfeng Wang** [2 5]

## Abstract

Addressing reward design complexities in deep reinforcement learning is facilitated by knowledge transfer across different domains. To this end, we define *reward translation* to describe the cross-domain reward transfer problem. However, current methods struggle with non-pairable and non-time-alignable incompatible MDPs. This paper presents an adaptable reward translation framework *neural reward translation* featuring *semi-alignable MDPs*, which allows efficient reward translation under relaxed constraints while handling the intricacies of incompatible MDPs. Given the inherent difficulty of directly mapping semi-alignable MDPs and transferring rewards, we introduce an indirect mapping method through reward machines, created using limited human input or LLM-based automated learning. Graph-matching techniques establish links between reward machines from distinct environments, thus enabling cross-domain reward translation within semi-alignable MDP settings. This broadens the applicability of DRL across multiple domains. Experiments substantiate our approach's effectiveness in tasks under environments with semi-alignable MDPs.

## 1. Introduction

Deep reinforcement learning (DRL) has made notable strides in areas such as gaming (Mnih et al., 2015; Lample and Chaplot, 2017), robotic control (Kober et al., 2013),

autonomous driving (Zhu et al., 2018), and precision agriculture management (Li et al., 2021). However, it is recognized that the effectiveness of current DRL methods hinges on the quality of the provided reward signals. Dense rewards require significant engineering effort in practical applications (Fickinger et al., 2021), and non-Markovian reward signals are often produced by humans (MacGlashan et al., 2017), potentially hindering the training process. In light of these challenges, it is important to consider how humans excel at drawing on their own experiences to learn and apply skills across various domains. Specifically, humans demonstrate a unique ability to recognize latent structural similarities between tasks in related but distinct areas, which enables them to abstract skills from differences. They adeptly learn from third-party observations that lack explicit correspondence to internal self-representations (Stadie et al., 2017; Liu et al., 2018; Sermanet et al., 2018) —performing tasks like finding a path on a map and navigating to a destination in real life—or imitating experts with different embodiments (Gupta et al., 2017; Rizzolatti and Craighero, 2004; Liu et al., 2019) in unfamiliar environments (Liu et al., 2019). Endowing RL agents with this ability and transfer rewards across different domains could greatly enhance their learning efficiency and reduce the efforts in reward design.

Existing efforts for transferring rewards across various domains primarily focus on cross-domain imitation learning, including three main types of domain discrepancies: dynamics (Liu et al., 2019), embodiment (Gupta et al., 2017; Hudson et al., 2022), and viewpoint mismatch (Jiang et al., 2020; Stadie et al., 2017). These approaches require paired and time-aligned demonstrations, often hard to handle more than one descriptor at once. Subsequent advances, such as DAIL (Kim et al., 2020) and other works using Cycle-GAN (Raychaudhuri et al., 2021) and Gromov-Wasserstein distance (Fickinger et al., 2021), relax some constraints, allowing algorithms to automatically learn observation and action mappings from unpaired and unaligned demonstrations. However, they still struggle with incompatible MDPs which are non-pairable and non-time-alignable tasks.

In this paper, we introduce the concept of *reward translation*, describing cross-domain reward transfer in reinforcement learning as a process akin to translation between different languages. We further introduce *semi-alignable MDPs*, which describe incompatible MDPs sharing an abstract

---

[*]Equal contribution  [1]Antai College of Economics and Management, Shanghai Jiao Tong University, Shanghai, China [2]School of Computer Science and Technology, East China Normal University, Shanghai, China [3]School of Software Engineering, Tongji University, Shanghai, China [4]School of Data Science, The Chinese University of Hong Kong, Shenzhen, Shenzhen, China [5]Key Laboratory of Mathematics and Engineering Applications, East China Normal University, Shanghai, China. Correspondence to: Xiangfeng Wang <xfwang@cs.ecnu.edu.cn>.

*Proceedings of the 42$^{nd}$ International Conference on Machine Learning*, Vancouver, Canada. PMLR 267, 2025. Copyright 2025 by the author(s).

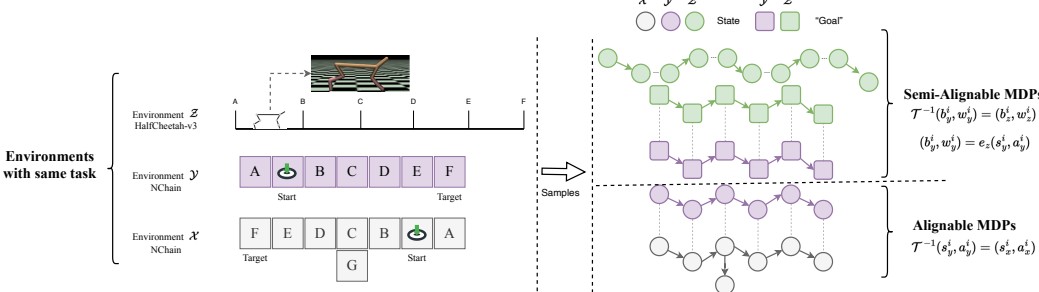

Figure 1. The distinction between alignable and semi-alignable MDPs involves injective mappings. In NChain environments and the HalfCheetah-v3, agents aim for point F. Alignable MDPs exhibit state and action space injections, seen in the NChain examples (Environment $\mathcal{Y}$ to $\mathcal{X}$). Semi-alignable MDPs, as in NChain-HalfCheetah pairings (Environment $\mathcal{Y}$ to $\mathcal{Z}$), lack these injections but share a common goal, enabling mappings between abstract "goal" and "skill" spaces.

alignment, as illustrated in Figure 1. By further abstracting tasks to focus on skills and sub-tasks, semi-alignable MDPs permit high-level mapping across domains, more similar to human learning processes. Our goal is to establish a framework capable of learning these high-level mappings within generalized semi-alignable MDPs and reusing rewards to enhance training efficiency in new domains. To achieve this, we must abstract tasks to obtain alignable skill and abstract spaces. Reward machines (RMs) provide a bridge to discover task structures with non-Markovian reward functions through the use of high-level events and abstract checkpoints (Icarte et al., 2022). We propose the *neural reward translation* (NRT) framework to tackle the reward transfer problem within semi-alignable MDPs. NRT leverages RMs to distill abstract alignments and transfer reward signals between domains.

The primary contributions of this paper are as follows:

1) The introduction of semi-alignable Markov decision processes (MDPs), providing a crucial theoretical foundation for facilitating reward translation in cross-domain reinforcement learning and extending beyond alignable MDPs.

2) The development of a novel framework called neural reward translation (NRT), designed to address the reward translation problem within semi-alignable MDPs by building upon the foundation provided by reward machines. Moreover, we incorporate an innovative reward machine generator that utilizes large language models for the automatic generation of reward machines.

3) The proposal of several semi-alignable environments [1], showcasing the effectiveness of the Neural Reward Translation approach in handling reward translation tasks where

environments operate under semi-aligned MDPs.

## 2. Related Works

**Domain transfer in RL**. Various works have conducted transfer learning in the reinforcement learning area (Taylor and Stone, 2009; Zhu et al., 2023). To deal with transfer learning between different domains, primitive methods always try to use the hand-craft features along with a distance metric between the imitation agent and the expert. For example, Ammar and Taylor (2011) established a shared state space between MDP in different domains and learns a map between states, while Ammar et al. (2015) applies unsupervised manifold alignment to learn linear maps between states possessing similar local geometric properties. Existing efforts for domain transferring primarily focus on cross-domain imitation learning, relying on three key domain descriptors: dynamics (Liu et al., 2019), embodiment (Gupta et al., 2017; Hudson et al., 2022), and viewpoint mismatch (Jiang et al., 2020; Stadie et al., 2017). Generally, these approaches acquire states corresponding to proxy tasks and paired, time-aligned demonstrations to learn a state map or state encoder using deep learning methods. DAIL (Kim et al., 2020) proposes a comprehensive framework encompassing all three types and employs GAMA to autonomously learn alignment between MDPs from distinct domains. Moreover, xDIO (Raychaudhuri et al., 2021) uses CycleGAN for alignment with state-only demonstrations, and GWIL (Fickinger et al., 2021) employs Gromov-Wasserstein distance to eliminate proxy task need. These domain transfer approaches assume alignable MDPs, which are often scarce in real-world applications. In contrast, this paper aims to distill abstract alignment from semi-alignable expert demonstrations and transfer it to the imitation agent through an additional reward signal, circumventing the need for alignable MDPs and contributing to a more general domain transfer in reinforcement learning.

---

[1]An early-stage version of the code is available at: https://github.com/hyyh28/reward_translation. Note that the codebase is still under development and may lack full documentation or polish.

**Reward Machine**. Reward machines (RM) were initially introduced as a class of finite state machines by (Icarte et al., 2018). These machines were designed to unveil the structure of non-Markovian reward functions in tasks represented by high-level events (i.e., propositional variables). Icarte et al. (2018) combined Q-learning with reward machines, proposing QRM, the first reinforcement learning method within RM. Subsequently, Icarte et al. (2022) introduced counterfactual experiences for reward machines (CRM), a modified version of QRM capable of learning a single Q-function that considers RM states as inputs, thus enhancing compatibility with deep neural networks. Icarte et al. (2022) proposed hierarchical reinforcement learning for reward machines (HRM) to accelerate policy learning, albeit possibly converging to sub-optimal solutions. Icarte et al. (2023) formulated a discrete optimization problem for experience-based learning of reward machines in partially observable environments. Reward machines have been applied to solve problems in robotics (Camacho et al., 2021; DeFazio and Zhang, 2021; Shah et al., 2020), multi-agent reinforcement learning (Neary et al., 2021), lifelong reinforcement learning (Zheng et al., 2022), self-paced reinforcement learning (Koprulu and Topcu, 2023) and offline reinforcement learning (Sun and Wu, 2023). Unlike these studies, our paper seeks to employ reward machines to uncover task structures in reinforcement learning and facilitate reward signal transfer between tasks from different domains.

**Reinforcement learning abstraction**. Our work is also related to reinforcement learning abstraction, which aims to improve the efficiency and scalability of reinforcement learning algorithms when dealing with complex tasks by creating abstract representations of environments or tasks (Abel, 2022). Various types of abstractions have been explored, including state (Zhang et al., 2020; Abel et al., 2019; Kamalaruban et al., 2020; Cortese et al., 2021), action (Şimşek et al., 2005; Machado et al., 2017), and temporal transition (Abel et al., 2020; Biza and Platt, 2019; Ma et al., 2021; Yang et al., 2021). Reward machines can also be considered as a reinforcement learning abstraction method. However, while these works mainly focus on creating abstract representations for a single environment or task to reduce the complexity of the state and action spaces, thus making the task easier to learn, our paper aims to use abstraction to achieve high-level knowledge transfer between different reinforcement learning environments.

## 3. Preliminaries

Before a comprehensive introduction of the neural reward translation (NRT) framework, it is pertinent to present an overview of the problem context concerning reward translation, outline the principles of semi-alignable MDPs as well as the introduction about reward machine, as this underpins the theoretical foundation.

### 3.1. Reward Translation

In this study, *reward translation* refers to cross-domain reward transfer in reinforcement learning. Formally, it involves translating rewards from a source domain $X$ to a target domain $Y$. Conventional cross-domain methods assume alignable MDPs, employing a formulation where state-action alignment between source and target domains is predefined:

$$r_y(s_y, a_y) = r_x(\mathcal{T}^{-1}(s_y, a_y)), \tag{1}$$

where $s$ denotes the state, $a$ represents the executed action, and $\mathcal{T}^{-1}$ is an inverse transformation function mapping elements in $Y$ back to their counterparts in $X$. This direct mapping approach relies strictly on original state-action pairs, rendering it ineffective for non-alignable MDPs lacking pairability and time-alignability. To address this limitation, we propose a soft version reward translation, which enables reward transfer based on embedded state-action representations:

$$r_y(e(s_y, a_y)) = r_x(\mathcal{T}^{-1}(e(s_y, a_y))), \tag{2}$$

where $e$ is an embedding function that projects state-action pairs from different domains into a shared latent space. Figure 1 compares these translation approaches, illustrating how high-level abstractions, "Goal" and "Skill," serve as embeddings for state and action, facilitating reward translation in MDPs without strict pairability and time-alignability. A detailed discussion of "Goal" and "Skill" follows in the next subsection.

### 3.2. Semi-Alignable MDPs

To delineate the soft reward translation problem formulated in Equation. (2), we present the notion of **semi-alignable MDPs** (Illustrated in Figure 1) to portray incompatible MDPs with concealed maps. In this paper, we elegantly introduce the notions of "goal" and "skill," symbolized as $b$ and $w$. These concepts encapsulate the essence of high-level states and actions, which can be succinctly conveyed through the equation $(b, w) = e(s, a)$. To offer a tangible interpretation, a "goal" may be likened to a series of checkpoints in games, representing coarse-grained states. In contrast, "skill" captures the nuances of incremental differences between each step's "goals." The infinite horizon Markov decision process (MDP) $\mathcal{M}$ can be augmented to be described as a tuple $\langle S, A, B, W, \Pr, \Pr^B, r \rangle$, where $S$ signifies the state space, $A$ represents the action space, $\Pr$ indicates the transition function based on state and action, while $\Pr^B$ denotes the deterministic transition function on "goal" and $r$ refers to the reward function. The infinite horizon Markov decision process (MDP) $\mathcal{M}$ can be augmented to be described as a tuple $\langle S, A, B, W, \Pr, \Pr^B, r \rangle$,

where $S$ signifies the state space, $A$ represents the action space, $B$ denotes the set of abstract states or "goals", $W$ represents the set of possible transitions between abstract states, Pr indicates the transition function based on state and action, $\text{Pr}^B$ denotes the deterministic transition function on abstract states, and $r$ refers to the reward function. Let $B = \{b^i\}_{i=0}^n$ and $W = \{w^i\}_{i=0}^n$ denote the extended "goal" space and "skill" space, respectively. The augmented MDP for the environment within domain $x$ can be defined as $\mathcal{M}_x^{\mathcal{T}} = (S_x, A_x, B_x, W_x, \text{Pr}_x, \text{Pr}_x^B, r_x^{\mathcal{T}})$. Additionally, we introduce an intelligible albeit less rigorous definition of semi-alignable MDPs. For the extended $\mathcal{M}_x^{\mathcal{T}}$ and $\mathcal{M}_y^{\mathcal{T}}$, if there exist injections between $(w_x^i \in W_x, b_x^i \in B_x)$ and $(w_y^i \in W_y, b_y^i \in B_y)$, they are deemed semi-alignable MDPs. While the precise definition of semi-alignable MDPs is an extension of the *MDP alignability theory* and the *MDP reduction* definition proposed by (Kim et al., 2020).

**Definition 3.1.** An **MDP semi-reduction** from $\mathcal{M}_x^{\mathcal{T}}$ to $\mathcal{M}_y^{\mathcal{T}}$, denoted as $\mathcal{M}_x^{\mathcal{T}} \succeq \mathcal{M}_y^{\mathcal{T}}$ can be represented by a tuple

$$r = (\phi, \psi), \quad \phi : B_x \to B_y, \psi : W_x \to W_y.$$

For all $(b_x, w_x, b_y, w_y) \in B_x \times W_x \times B_y \times W_y$ we have

1). $w$-optimality:

$$\begin{cases} O_{M_y}(\phi(b_x), \psi(w_x)) = 1 \Rightarrow O_{M_x}(b_x, w_x) = 1, \\ O_{M_y}(b_y, w_y) = 1 \Rightarrow \phi^{-1}(b_y) \neq \varnothing, \psi^{-1}(w_y) \neq \varnothing, \end{cases}$$

where $O_M(b, w) = 1$ implies that $(b, w)$ exists in $M$ and is produced by the optimal policy;

2). $b$-dynamic: if $O_{M_y}(b_y, w_y) = 1$, we have

$$\text{Pr}_y^B(b_y, w_y) = \phi\left(\text{Pr}_x^B(b_x, w_x)\right), b_x \in \phi^{-1}(b_y), w_x \in \psi^{-1}(w_y).$$

By distilling "goals" and "skills" for optimal policies $M_x$ and $M_y$, a graph of goal-skill trajectories emerges. Figure 3.2 displays this with HalfCheetah-v3 and NChain instances. When MDP semi-reduction is present, homomorphic trajectories reveal underlying structural similarities between $M_x$ and $M_y$. We then define semi-alignable:

**Definition 3.2.** Two MDPs $M_x$ and $M_y$ are semi-alignable if and only if $M_x \succeq M_y$ or $M_y \succeq M_x$, where $M_x$ and $M_y$ are any two MDPs in domain $\mathcal{X}$ and domain $\mathcal{Y}$, while $M_x \succeq M_y$ conveys that there exists a tuple $(\phi, \psi)$ enabling semi-reduction from $M_x$ to $M_y$.

Efficiency in achieving the target within (2) improves after applying the semi-reduction definition. However, accessing abstract "goal" and "skill" directly is challenging, making semi-reduction establishment difficult. Therefore, we utilize the *reward machine* (RM) to indirectly link these two spaces.

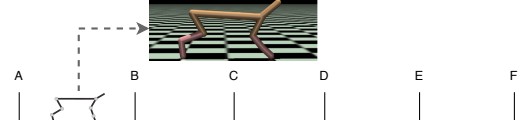

(a) HalfCheetah-v3 task. The target of the robot is to reach F.

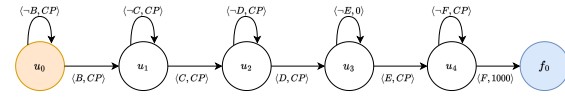

(b) Reward machine for HalfCheetah-v3 task. $f_0 \in F$ denotes the terminal state.

*Figure 2.* Example of reward machine for HalfCheetah-v3. The reward machine is designed by (Icarte et al., 2022).

### 3.3. Reward Machine

The formulation of reward machines, which are designed to unveil the organization of non-Markovian reward functions related to tasks characterized by high-level events. Reward machines is typically defined as (Icarte et al., 2018):

**Definition 3.3.** (Reward Machine). Given a set of propositional symbols $\mathcal{P}$, a set of (environment) states $\mathcal{S}$, and an action domain $x$, a finite state reward machine (RM) is defined as a tuple $\mathcal{R}_{PSA} = \langle U, u_0, F, \mathcal{P}, \delta_u, \delta_r \rangle$, where: $U \subseteq \mathcal{S}$ represents a finite collection of states, $u_0 \in U$ denotes the initial state, $F$ defines a restricted subset of terminal states ($F \cap U = \varnothing$), $\mathcal{P}$ signifies the set of propositional symbols, $\delta_u$ characterizes the state-transition function $\delta_u : U \to [U \times P \to U]$, and $\delta_r$ embodies the state-reward function $\delta_r : U \to [U \times P \times U \to \mathbb{R}]$.

It should be emphasized that employing a reward machine in reinforcement learning necessitates the extension of the fundamental MDP.

**Definition 3.4.** (MDP with Reward Machine). An MDP integrated with a Reward Machine is represented as a tuple $\mathcal{T}_{\mathcal{R}_{PSA}} = \langle S, A, \text{Pr}, \gamma, P, L, U, u_0, \delta_u, \delta_r \rangle$, where $S, A, \text{Pr}, \gamma$ pertain to the state space, action space, transition function, and discount factor in the original MDP, while $P, U, u_0, \delta_u, \delta_r$ are determined by the RM. Additionally, $L$ [2] symbolizes a labelling function $L : S \times A \times S \to 2^{\mathcal{P}}$.

In each step, the agent executes action $a$ in the MDP, transitioning from state $s$ to $s'$, while the RM changes to state $u' = \delta_u(u, L(s'))$, rewarding $\hat{r}(s, a, s', u, u')$. Figure 2 illustrates an RM example for HalfCheetah, where the robot aims to reach target $F$ within a set number of steps. Unlike RL environmental states ($s$), RMs possess states ($u$) and transitions depending on $\mathcal{P}$, denoting the robot's relative

---

[2]$L$ is a labeling function that assigns to each transition $(s, a, s')$ a set of atomic propositions $\mathcal{P}$, indicating which events are considered to have occurred during that transition.

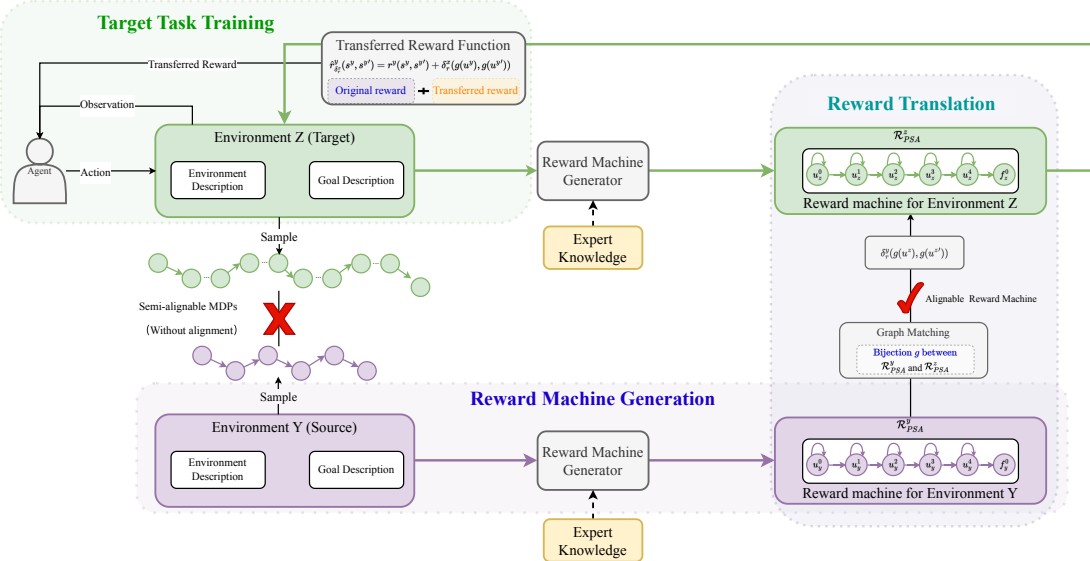

*Figure 3.* An overview of the Neural Reward Translation (NRT) framework: 1) Reward machine generation uses a generator to construct RMs based on task and environment descriptions through expert design. 2) Reward translation aligns source and target RMs using graph matching, facilitating reward transfer. 3) Target task training leverages transferred rewards for efficient learning.

position to the target. The path has six checkpoints; crossing one causes the RM state to advance. RMs decompose RL problems, providing coarse-grained disassembly to inform generalized cross-domain knowledge, crucial due to significant differences among diverse tasks in fine-grained states, actions, and transitions because such coarse-grained disassembly will provide a channel to indirectly access to the "goal" and "skill" within semi-alignable MDPs. Ultimately, the reward machine state space $U$ materializes as an injective mapping originating from the agent state space $S$. We can now propose the following theorem to establish a rigorous connection between "goal" and "skill" with RMs:

**Theorem 3.5.** *Given a reward machine $\mathcal{R}^x_{PSA}$ for an environment $x$, it is posited that a "goal" space $B_x$ and a "skill" space $W_x$ exist. Furthermore, injective mappings $\Theta_x : U_x \rightarrow B_x$ and $\Gamma_x : P_x \rightarrow W_x$ are also postulated to exist. Under these stipulations, the ensuing relationships emerge:*

*1) For any "goal" $b^i_x \in B_x$, there exists a unique reward machine state $u^i_x \in U$ or $f^i_x \in F_x$ such that $\Theta_x(u^i_x) = b^i_x$ or $\Theta_x(u^i_x) = f^i_x$;*

*2) For any "skill" $w^i_x \in W^x_x$, there exists a unique propositional symbol $p^i_x \in P_x$ such that $\Gamma_x(p^i_x) = w^i_x$.*

A comprehensive proof is in Appendix A.1. This theorem enables recasting the reward translation problem into the more tractable reward machine transfer problem. Each reward machine, represented as a finite state machine by a graph, transforms the task into a graph matching problem for transfer between two reward machines.

## 4. Reward Translation via Reward Machine

In this section, we introduce the primary framework, *Neural Reward Translation* (NRT), devised to address the proposed reward translation problem between reward machines. The NRT framework comprises three core components: 1) reward machine generation component; 2) reward translation component; and 3) target task training component. The Reward Machine (RM) generation component constructs RMs based on task and environment descriptions, leveraging expert knowledge and an RM Generator. This study also integrates a Large Language Model (LLM) for automated RM construction. The reward translation component employs standard graph matching algorithms to align generated RMs from the source task to the target task to transfer reward. During target task training, the agent learns within the target environment, leveraging transferred rewards from the source task to enhance learning efficiency.

**Reward Machine Generation**. The primary role of this component is to construct reward machines by incorporating domain knowledge. Typically, experts manually design reward machines based on environment and target information (Icarte et al., 2018). In this work, we also propose a method to construct reward machines using large language models (LLMs). However, since this is not the core focus of our study, we provide only a brief discussion.

The LLM is supplied with environment, observation, action, and target descriptions, detailing the task and domain alongside the definition of a reward machine. To facilitate few-shot learning in new tasks, examples of reward machine

generation are also included. Due to the complexity of learning reward machines, the LLM must consider various aspects such as the design of propositional symbols, the injection between the agent's observations and RM states, and the RM transitions. We utilize a chain of thought (COT) approach when designing the training prompt for the LLM. Generally, our prompt sequentially poses three questions to the LLM, enabling the development of an RM structure for a given environment: i) Design the set of propositional symbols $\mathcal{P}$ for $\langle \text{ENV} \rangle$ environment; ii) Design the get event function for $\langle \text{ENV} \rangle$ environment; iii) Design the reward machine for $\langle \text{ENV} \rangle$ environment.

Firstly, to design the set of propositional symbols $\mathcal{P}$, the LLM needs to identify key events within the task description and utilize them as a portion of the propositional symbols. Next, in order to design the get event function, the LLM must establish a mapping between events that are accessible in the environment code and the propositional symbols. Subsequently, Python code is generate by LLM, enabling the agent to access reward machine events and states. Finally, to construct the reward machine, the LLM reviews the propositional symbols and extracts general states $u \in \mathcal{U}$ as well as terminal states $f \in \mathcal{F}$. The LLM formulates reward machine transitions, including sparse rewards exclusively present upon task completion. Detailed prompts can be found in Appendix A.3.

However, the generated or expert designed reward machine only contains sparse rewards; therefore, NRT introduces a denser reward for the source task's reward machine. At each stage, the agent operates within the environment and executes action $a$, transitioning from state $s$ to $s'$ in the MDP. The RM progresses from state $u$ to $u' = \delta_u(u, L(s'))$, and the agent acquires a reward comprised of the original reward and the reward from the RM:

$$\hat{r}(s, a, s', u, u') = r(s, a, s') + \delta_r(u, u'), u' = \delta_u(u, L(s')). \tag{3}$$

$\delta_r$ is depicted using a potential-based reward shaping framework to maintain policy consistency:

$$\delta_u(u, u') = \gamma_{\mathcal{R}} \phi(u') - \phi(u), \tag{4}$$

where $\gamma_{\mathcal{R}}$ represents the discount parameter. Moreover, to compute the potential value of the corresponding reward machine state, NRT incorporates the expert of optimal value function of the agent in source task $g$ as follows:

$$\phi(u) = \mathbb{E}_{s_i \sim S} \left[ V^* \left( s_i, u_i \mid u_i = u \right) \right]. \tag{5}$$

Consequently, RMs for both source and target tasks are generated, while the obtained reward machine for the source task possesses a dense reward derived from its learned optimal value function. The subsequent challenge involves the transfer of dense rewards from the source task to target task.

**Reward Translation**. The focus of this component is to achieve effective reward translation between source and target tasks within distinct domains under semi-alignable Markov Decision Processes (MDPs). Initially, during the reward machine generation component, the Neural Reward Translation (NRT) method constructs reward machines for both source and target tasks. And the reward machine for the source task incorporates a dense reward obtained from its learned optimal value function.

Subsequently, this component transfers the dense reward originating from the source task's reward machine to the target task's reward machine. As the reward machines are aligned via graph matching, NRT defines two types of relationships between the reward machines and seeks to accomplish reward translation based on these relationships. Definitions for isomorphic and homomorphic reward machines are presented accordingly (In the following of this paper, we use $g$ and $h$ to represent the map between reward machine states).

**Definition 4.1.** (Isomorphic reward machine). Let $\mathcal{R}^x_{PSA}$ and $\mathcal{R}^y_{PSA}$ denote two reward machines. $\mathcal{R}^x_{PSA}$ is a isomorphic reward machine to $\mathcal{R}^y_{PSA}$ if and only if there exist bijections $h : \mathcal{P}_x \to \mathcal{P}_y$ and $g : \mathcal{U}_x \to \mathcal{U}_y$ such that $p^i_y = h(p^i_x), p^i_x = h^{-1}(p^i_y)$ and $u^i_y = g(u^i_x), u^i_x = g^{-1}(u^i_y)$.

Considering the priorly introduced bijections $b_i = \Theta(u_i)$ and $p_i \in P$: $w_i = \Gamma(p_i)$ in Theorem. 3.5, Theorem 4.2 is utilized to characterize the correlation among semi-alignable MDPs with isomorphic reward machines.

**Theorem 4.2.** *If $M_x$ and $M_y$ are two MDPs satisfying $M_x \succeq M_y$ and $M_y \succeq M_x$, then their tasks possess isomorphic reward machines $\mathcal{R}^x_{PSA}$ and $\mathcal{R}^y_{PSA}$.*

The comprehensive proof can be found in Appendix A.1. For MDPs $M_x$ and $M_y$, should isomorphic reward machines exist for their tasks, a reinforcement learning agent's reward function in $M_y$ can be represented as:

$$\hat{r}^{\delta^x_r}_y(s_y, a_y, s_y{'}, u_y, u_y{'}) = r_y(s_y, a_y, s_y{'}) + \delta^y_r(u_y, u_y{'}), \tag{6}$$

and if $M_y$ employs the transferred $\delta^x_r$, the reward function will be indicated by:

$$\hat{r}^{\delta^x_r}_y(s_y, a_y, s_y{'}, u_y, u_y{'}) = r_y(s_y, a_y, s_y{'}) + \delta^x_r(g(u_y), g(u_y{'})). \tag{7}$$

Given that both $\delta^x_r$ and $\delta^y_r$ employ potential-based reward shaping, we can deduce the following equation by examining the following potential-based reward shaping properties:

$$\sum_{i=0}^{T-1} \hat{r}^{\delta^y_r}_y(x^i_y) = \sum_{i=0}^{T-1} \hat{r}^{\delta^x_r}_y(x^i_y), \tag{8}$$

where $T$ represents the task duration, and $x^i_y$ donates $\langle s^i_y, a^i_y, s^{i+1}_y, u^i_y, u^{i+1}_y \rangle$. The above relations also hold for

transferring state reward functions from $M_y$ to $M_x$. However, isomorphic reward machines impose restrictive conditions demanding a bijection between semi-alignable MDPs' reward machines. Therefore, homomorphic reward machines are introduced as a more flexible formulation.

**Definition 4.3.** (Homomorphic reward machine). Let $\mathcal{R}_{PSA}^x$ and $\mathcal{R}_{PSA}^y$ denote two reward machines. $\mathcal{R}_{PSA}^x$ is a homomorphic reward machine to $\mathcal{R}_{PSA}^y$ if and only if there exist injections $h : \mathcal{P}_x \to \mathcal{P}_y$ and $g : \mathcal{U}_\S \to \mathcal{U}_\dagger$ such that $p_y^i = h(p_x^i)$ and $u_y^i = g(u_x^i)$.

Additionally, considering the bijections $b_i = \Theta(u_i)$ and $p_i \in P$: $w_i = \Gamma(p_i)$ introduced in Theorem 3.5, Theorem 4.4 is utilized to characterize the correlation among semi-alignable MDPs with homomorphic reward machines.

**Theorem 4.4.** *If $M_x$ and $M_y$ are two MDPs satisfying either $M_y \succeq M_x$ or $M_x \succeq M_y$, then their tasks possess homomorphic reward machines $\mathcal{R}_{PSA}^x$ and $\mathcal{R}_{PSA}^y$.*

The comprehensive proof can be found in Appendix A.1. For MDPs $M_x$ and $M_y$, if $M_x \succeq M_y$, the reinforcement learning agent's reward function in $M_y$ is denoted as in Equation (6). When $M_y$ employs the transferred state reward functions from the reward machine of $M_x$, the reward function remains consistent with Equation (7) while maintaining the characteristics described in Equation (8). Nevertheless, transferring rewards from $M_y$ to $M_x$ necessitates the development of a piecewise function as depicted in Equation (9), due to the absence of a bijection.

$$\hat{r}_x^{\delta_r^y}(s_x, a_x, s_x{}', u_x, u_x{}') = r_x(s_x, a_x, s_x{}') +$$
$$\begin{cases} \delta_r^y(u_y, u_y{}'), & \text{if } u_x = g(u_y) \text{ and } u_x{}' = g(u_y{}'), \\ \delta_r^x(u_x, u_x{}'), & \text{otherwise.} \end{cases} \quad (9)$$

As a result, a reward translation channel is established between semi-alignable MDPs. To transfer the reward function, corresponding reward machines are first constructed, and subsequently, the state reward function for the original task is computed using Equation 5.

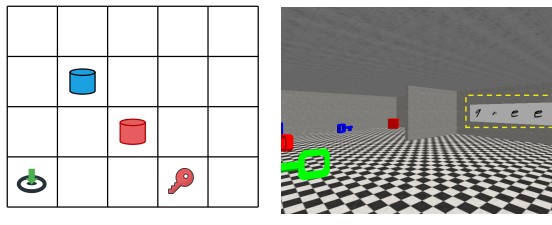

(a) Original task: Text-Sign (b) Target task: Miniworld-Sign

*Figure 4.* The Sign task for the 3D Visual Navigation environment.

**Target Task Training**. The primary goal of this component is to instruct the target task by utilizing the transferred re-

ward received from the source task within a distinct domain. In the stage, the agent operating in the target task has access to the transferred reward function. The agent acquires an observation $s_y{}'$ from the target task environment at every step of training. This observation is subsequently fed into the reward machine, which yields the corresponding reward machine state $u_y{}'$. Given the observation and reward machine state $(s_y, a_y, s_y{}', u_y, u_y{}')$, the agent acquires the transferred reward $\hat{r}_y^{\delta_r^x}(s_y, a_y, s_y{}', u_y, u_y{}')$ from the transferred reward function. Subsequently, the agent can employ diverse reinforcement learning algorithms to train its policy, aiming to maximize the aggregate transferred reward:

$$J(\theta) = \mathbb{E}_{\tau_y \sim p_\theta}\left[\sum_{i=0}^{T-1} \hat{r}_y^{\delta_r^x}(s_y^i, a_y^i, s_y^{i+1}, u_y^i, u_y^{i+1})\right], \tag{10}$$

where $J(\theta)$ denotes the objective function dependent on the policy parameters $\theta$, and $T$ represents the time horizon.

## 5. Experiments

This section presents experiments addressing two primary research questions: 1) Can the NRT framework effectively extract abstract alignments from semi-alignable MDPs across reinforcement learning domains? 2) Does transferred reward, based on abstract alignment, enhance efficiency and performance in target task training? To investigate these questions, we evaluate the NRT framework in two sparse reward settings: 3D Visual Navigation and MuJoCo, verifying both isomorphic and homomorphic reward machines. We use the state-of-the-art Proximal Policy Optimization (PPO) algorithm as the baseline and conduct an ablation study with three variants: PPO-RM, PPO-NRT(Reward), and PPO-NRT(RM+Reward). PPO-RM augments the agent's observation with the reward machine state but does not transfer rewards. PPO-NRT(Reward) transfers rewards without incorporating the reward machine state in the observation. PPO-NRT(RM+Reward) integrates both reward transfer and reward machine state information. Due to space constraints, detailed experimental descriptions are in Appendix A.4.

### 5.1. 3D Visual Navigation

In the 3D visual navigation environment, we selected the Sign task in the Miniworld (Chevalier-Boisvert et al., 2023) as the target task, with the Text-Sign task serving as the original task. Figures. 4 depict the original and target tasks, respectively. In the Miniworld-Sign environment, the agent receive a sparse reward only if it reaches all the targets including the red key, the blue box and the red box. Figure. 5(a) illustrates the reward transfer between the isomorphic reward machines in the Text-Sign and Miniworld-Sign tasks. From Figure. 5(b), our results show that in the Miniworld-Sign environment, where rewards are sparse, both PPO and PPO-RM—despite using original or extended observations

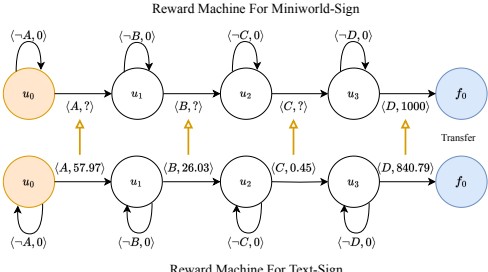

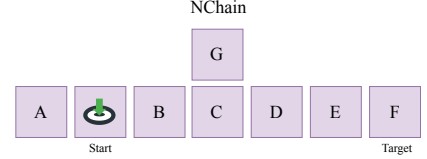

(a) Original task: NChain

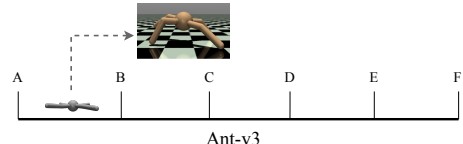

(b) Target task: Mujoco-Ant

*Figure 6.* The Ant task in Mujcoco environments.

(a) Text-Sign to Miniworld-Sign

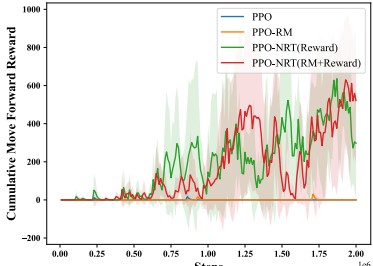

(b) Learning curve for Miniworld-Sign

*Figure 5.* Experiment for the 3D Visual Navigation environment. (a) The isomorphic reward machines and the cross domain reward transfer process for Text-Sign and Miniworld-Sign; (b) The learning curve for Miniworld-Sign environment.

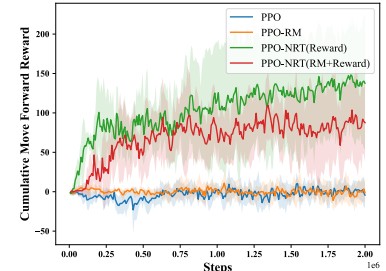

(a) Learning curve for Mujoco-Ant

with reward machine information—struggle to overcome the sparse reward issue. In contrast, PPO-NRT(Reward) and PPO-NRT(RM+Reward), which leverage transferred rewards, significantly alleviate the problem and achieve superior performance. These findings highlight the effectiveness of the NRT framework in extracting alignments from semi-alignable MDPs and improving training through the transfer of rewards within isomorphic reward machines.

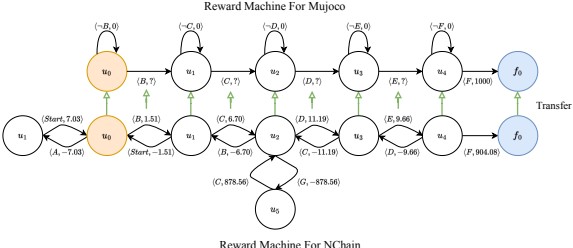

(b) NChain to Mujoco

### 5.2. Mujoco Experiment

We conducted experiments using Mujoco environments, selecting HalfCheetah, Hopper, and Ant as target tasks, with a simplified NChain game as the source task. Due to space constraints, we present only MuJoCo-Ant results in the main text, while the full results are included in the Appendix. A.4. Figure 12 in the Appendix illustrates the experimental environments. All target tasks follow the standard OpenAI Gym settings (Brockman et al., 2016) but employ sparse rewards, where the agent receives $r = 1000$ only upon reaching predefined goals. The source task involves navigating a linear state sequence to reach point F, earning a sparse reward. Figure 6 presents the original and target tasks in MuJoCo-Ant training. Figure 7(b) depicts the generated reward machine and the cross-domain reward transfer process from NChain to MuJoCo, applicable to all three target

*Figure 7.* Experiment for the Mujoco-Ant environment. (a)The homomorphic reward machine and the cross-domain reward transfer process of NChain to Mujoco. (b) The learning curve for Ant task.

tasks. The reward machines for the source and target tasks exhibit homomorphism, enabling the source task's reward signals to guide learning in the target environments. As shown in Figure 7(a), our results indicate that in extremely sparse reward settings, augmenting observations with reward machine information alone is insufficient. However, leveraging transferred rewards substantially mitigates the sparsity issue, significantly improving learning efficiency. Notably, in Ant-v3 tasks, PPO-NRT(Reward) achieves the best performance.

# 6. Conclusion

In conclusion, this paper introduced the concept of *semi-alignable MDPs* alongside the *Neural Reward Translation* (NRT) framework to facilitate *reward translation* in reinforcement learning to reduce reward design complexities. NRT employs reward machines to address reward translation challenges within semi-alignable MDPs and features an innovative large language model-based generator for the automatic generation of reward machines. Our method significantly enhances training efficiency across various environments. Although challenges persist in constructing appropriate reward machines and deciphering relationships in complex tasks, future research endeavors will continue to explore the vast potential of semi-alignable MDPs and work towards broadening NRT's applicability in a diverse range of situations and domains.

## Impact Statement

This paper presents work whose goal is to advance the field of Machine Learning. There are many potential societal consequences of our work, none which we feel must be specifically highlighted here.

## Acknowledgement

This work was supported by the National Natural Science Foundation of China (NSFC, No. 62231019, 62406270, 72394361, 62106213), the Shanghai Science and Technology Committee (STCSM, No. 24YF2748800), and an extended support project from the Shenzhen Science and Technology Program.

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

# A. Appendix

## A.1. Proof

**Theorem.**3.5 *Given a reward machine $\mathcal{R}^x_{PSA}$ for an environment $x$, it is posited that a "goal" space $B_x$ and a "skill" space $W_x$ exist. Furthermore, injective mappings $\Theta_x : U_x \to B_x$ and $\Gamma_x : P_x \to W_x$ are also postulated to exist. Under these stipulations, the ensuing relationships emerge:*

*1) For any "goal" $b^i_x \in B_x$, there exists a unique reward machine state $u^i_x \in U_x$ or $f^i_x \in F_x$ such that $\Theta_x(u^i_x) = b^i_x$ or $\Theta_x(u^i_x) = f^i_x$;*

*2) For any "skill" $w^i_x \in W^x$, there exists a unique propositional symbol $p^i_x \in P_x$ such that $\Gamma_x(p^i_x) = w^i_x$.*

*Proof.* To prove the theorem, we need to show that the reward machine state space $U_x$ is a valid "goal" space $B_x$. Given the definitions of "goal" $b^i_x$, "skill" $w^i_x$, and reward machine, we observe that goals and skills are embedded in state-action pairs as: $(b^i_x, w^i_x) = e_x(s^i_x, a^i_x)$,

with $w^i_x$ being the difference between $b^i_x$ and $b^{i-1}_x$. Since the labeling function $L_x : S_x \times A_x \times S_x \to 2^{\mathcal{P}_x}$ represents the transition of $u_x$, we have:

$$u^i_x = L_x(s^i_x, a^i_x, s^{i+1}_x),$$

$$s^{i+1}_x = \mathrm{Tr}(s^i_x, a^i_x),$$

so we can get:

$$u^i_x = L_x(s^i_x, a^i_x).$$

By viewing reward machine state $u_x$ as "goal" $b_x$ and the reward state difference as defined "skills", the labeling function $L_x$ becomes an embedding $e_x$ for environment $x$. Hence, we demonstrate that reward machine state space $U_x$ is a valid "goal" space $B_x$.

$\square$

**Theorem. 4.2** *If two MDPs $M_x$ and $M_y$, $M_x \succeq M_y$ and $M_y \succeq M_x$, then their tasks have isomorphic reward machine $\mathcal{R}^x_{PSA}$ and $\mathcal{R}^y_{PSA}$.*

*Proof. Theorem 4.2* can be proven by showing both necessity and sufficiency.

**1. Proof of necessity:** If $M_x \succeq M_y$ and $M_y \succeq M_x$, then their tasks have isomorphic reward machines.

Let there exist semi-reductions $r_{x \to y} = (\phi_{x \to y}, \psi_{x \to y})$ from $M_x$ to $M_y$ and $r_{y \to x} = (\phi_{x \to y}, \psi_{y \to x})$ from $M_y$ to $M_x$. According to Definition 3.1, these semi-reductions satisfy the $\pi^w$-optimality and $y$-dynamic conditions for all beliefs and world-states.

We now construct bijections $h : P_x \to P_y$ and $g : \mathcal{U}_x \to \mathcal{U}_y$ as follows:

$$h(p^i_x) = \Gamma^{-1}_y(\phi_{x \to y}(w^i_x)) = \Gamma^{-1}_y(w^i_y) = p^i_y,$$

$$g(u^i_x) = \Theta^{-1}_y(\psi_{x \to y}(b^i_x)) = \Theta^{-1}_y(b^i_y) = u^i_y.$$

Since $\phi_{x \to y}$ and $\psi_{x \to y}$ are functions, so are their inverses. Besides, and $\Gamma$ and $\Theta$ are bijections. According to the transitivity of bijections. These bijections also satisfy the conditions of the isomorphic reward machines in Theorem 4.1, as both semi-reductions preserve the $\pi^w$-optimality and $y$-dynamic properties.

**2. Proof of sufficiency:** If the tasks have isomorphic reward machines, then $M_x \succeq M_y$ and $M_y \succeq M_x$.

Given the isomorphic reward machines $\mathcal{R}^x_{PSA}$ and $\mathcal{R}^y_{PSA}$ with established bijections $h : P_x \to P_y$ and $g : \mathcal{U}_x \to \mathcal{U}_y$, we will prove that $M_x \succeq M_y$ and $M_y \succeq M_x$.

Define mapping functions $\phi_{x \to y}(w^i_x) = \Gamma_y(h(p^i_x))$ and $\psi_{x \to y}(b^i_x) = \Theta_y(g(u^i_x))$. Since the task structures are isomorphic, these mappings can be used to construct the semi-reduction $(\phi_{x \to y}, \psi_{x \to y})$ from $M_x$ to $M_y$ that satisfies the $\pi^w$-optimality and $y$-dynamic conditions in Definition 3.1, thus showing that $M_x \succeq M_y$.

Similarly, use the inverse of given bijections for the semi-reduction $r_{y \to x} = (\phi_{y \to x}, \psi_{y \to x})$. That is, set $\phi_{y \to x}(w_y^i) = \Gamma_x(h^{-1}(p_y^i)) = \Gamma_x(p_x^i) = w_x^i$ and $\psi_{y \to x}(b_y^i) = \Theta_x(g^{-1}(u_y^i)) = \Theta_x(u_x^i) = b_x^i$. Since $h$ and $g$ are bijections, their inverses exist and are also bijections. By using these mappings, we can construct a semi-reduction from $M_y$ to $M_x$ that satisfies the conditions in Definition 3.1, showing that $M_y \succeq M_x$.

Having proven both necessity and sufficiency, we conclude the proof of Theorem 4.2. □

**Theorem. 4.4** *If two MDPs $M_x$ and $M_y$, $M_y \succeq M_x$ or $M_x \succeq M_y$, then their tasks have homomorphic reward machine $\mathcal{R}_{PSA}^x$ and $\mathcal{R}_{PSA}^y$.*

*Proof.* To prove Theorem 4.4, we need to show both necessity and sufficiency:

**1. Proof of necessity:** If $M_y \succeq M_x$ or $M_x \succeq M_y$, then their tasks have homomorphic reward machines.

Assume $M_x \succeq M_y$. Let there exist a semi-reduction $r_{x \to y} = (\phi_{x \to y}, \psi_{x \to y})$ from $M_x$ to $M_y$. According to Definition 3.1, these semi-reductions satisfy the $\pi^w$-optimality and $B$-dynamic conditions for all beliefs and world-states.

We now construct injection $h : \mathcal{P}_x \to \mathcal{P}_y$ and $g : \mathcal{U}_x \to \mathcal{U}_y$ as follows:

$$h(p_x^i) = \Gamma_y^{-1}(\phi_{x \to y}(w_x^i)) = \Gamma_y^{-1}(w_y^i) = p_y^i,$$
$$g(u_x^i) = \Theta_y^{-1}(\psi_{x \to y}(b_x^i)) = \Theta_y^{-1}(b_y^i) = u_y^i.$$

Since $\phi_{x \to y}$ and $\psi_{x \to y}$ are functions, so are their inverses. Besides, and $\Gamma$ and $\Theta$ are bijections. According to the transitivity of bijections. These bijections also satisfy the conditions the homomorphic reward machines in Theorem 4.3 because semi-reductions preserve the $\pi^w$-optimality and $B$-dynamic properties.

**2. Proof of sufficiency:** If the tasks have homomorphic reward machines, then $M_y \succeq M_x$ or $M_x \succeq M_y$.

Given the homomorphic reward machines $\mathcal{R}_{PSA}^x$ and $\mathcal{R}_{PSA}^y$ that satisfy the established injections $h : \mathcal{P}^x \to \mathcal{P}^y$ and $g : \mathcal{U}_x \to \mathcal{U}_y$, we will prove that either $M_y \succeq M_x$ or $M_x \succeq M_y$.

Assume without loss of generality that $|\mathcal{P}_y| \leq |\mathcal{P}_x|$ and $|\mathcal{U}_\dagger| \leq |\mathcal{U}_\S|$. Then we can define mapping functions $\phi_{x \to y}(w_x^i) = \Gamma_y(h(p_x^i)) = \Gamma_y(p_y^i) = w_y^i$ for every element in the domain of $h$. Similarly, define $\psi_{x \to y}(b_x^i) = \Theta_y(g(u_x^i)) = \Theta_y(u_y^i) = b_y^i$ for all elements in the domain of $g$

These mappings can be used to construct a semi-reduction $(\phi_{x \to y}, \psi_{x \to y})$ from $M_x$ to $M_y$ that satisfies the $\pi^w$-optimality and $B$-dynamic conditions in Definition 3.1. Thus, $M_x \succeq M_y$.

Having proven both necessity and sufficiency, we conclude that if two MDPs $M_x$ and $M_y$, $M_y \succeq M_x$ or $M_x \succeq M_y$, then their tasks have homomorphic reward machine $\mathcal{R}_{PSA}^x$ and $\mathcal{R}_{PSA}^y$. This completes the proof of Theorem 4.4. □

## A.2. Experiments

### A.2.1. FRAMEWORK

In this section, we present a comprehensive methodology for generating reward machines using a large language model (LLM), in our case, GPT-4, informed by domain-specific knowledge such as task manuals. As illustrated in Figure 8, we employ a few-shot learning strategy to familiarize the language model with reward machine design principles. This process involves defining the sets of propositional symbols $\mathcal{P}$ and reward machine states $\mathcal{U}$, creating an event extraction function that integrates the reward machine with the environment, and developing the transition function $\delta_u$ as well as the state reward function $\delta_r$.

Next, we equip the LLM with domain knowledge, which comprises both reward machine definitions and environmental descriptions sourced from the task manual. The LLM then answers user-generated queries using its acquired knowledge to assist in the creation of the reward machine. Due to the complex nature of reward machine development, we implement a chain of thought framework to bolster the LLM's reasoning capabilities.

### A.2.2. HAND-DEFINED RM FOR NCHAIN GAME

$$0 \quad \text{\# initial state}$$
$$[5] \quad \text{\# terminal state}$$
$$(0, 0,' !a!c', \text{ConstantRewardFunction}(-0.67))$$
$$(0, 1,' a', \text{ConstantRewardFunction}(-0.02))$$
$$(0, 3,' c', \text{ConstantRewardFunction}(-1.29))$$
$$(1, 1,' !b!g', \text{ConstantRewardFunction}(-0.75))$$
$$(1, 2,' b', \text{ConstantRewardFunction}(0.01))$$
$$(1, 0,' g', \text{ConstantRewardFunction}(-1.4))$$
$$(2, 2,' !a!grab', \text{ConstantRewardFunction}(-0.83))$$
$$(2, 5,' grab', \text{ConstantRewardFunction}(-7.48))$$
$$(2, 1,' a', \text{ConstantRewardFunction}(-1.58))$$
$$(3, 3,' !d!g', \text{ConstantRewardFunction}(-0.61))$$
$$(3, 4,' d', \text{ConstantRewardFunction}(-1.03))$$
$$(3, 0,' g', \text{ConstantRewardFunction}(0.01))$$
$$(4, 4,' !c', \text{ConstantRewardFunction}(-0.56))$$
$$(4, 3,' c', \text{ConstantRewardFunction}(-0.14))$$

### A.2.3. EXAMPLE OF LEARNING RM BY LLM

**Reward Machine Description:**

You are familiar with automata theory. A reward machine is defined as following:

Given a set of propositional symbols $\mathcal{P}$, a set of (environment) states $S$, and a set of actions $A$, a reward machine (RM) is a tuple $R_{PSA} = \ <U, u_0, F, \delta_u, \delta_r>$, where $U$ is a finite set of states, $u_0 \in U$ is an initial state, $F$ is a finite set of terminal states (where $U \cap F = \varnothing$, terminal states are not existed in $U$), $\delta_u$ is the state-transition function, $U \times 2^{\mathcal{P}} \to U \cup F$, $\delta_r$ is the reward-transition function, $U \to [U \times U \to \mathcal{R}]$

A reward machine $R_{PSA}$ starts in state $u_0$, and at each subsequent time is in some state $u_t \in U \cup F$. At every step $t$, the machine receives as input a truth assignment $\sigma_t$, which is a set that contains exactly those propositions in $\mathcal{P}$ that are currently true in the environment. For example, in an open door and get the key game, $\sigma_t = \{e\}$ if the agent opens the door $e$, while $\sigma_t = \{k\}$ if the agent gets the key $k$. Then the machine moves to the next state $u_{t+1} = \delta_u(u_t, \sigma_t)$ according to the state-transition function, and outputs a reward function $r_t = \delta_r(u_t)$ according to the state-reward function. This process repeats until the machine reaches a terminal state. Note that reward machines can model never-ending tasks by defining $F = \varnothing$.

## A.3. Learning RM by LLM

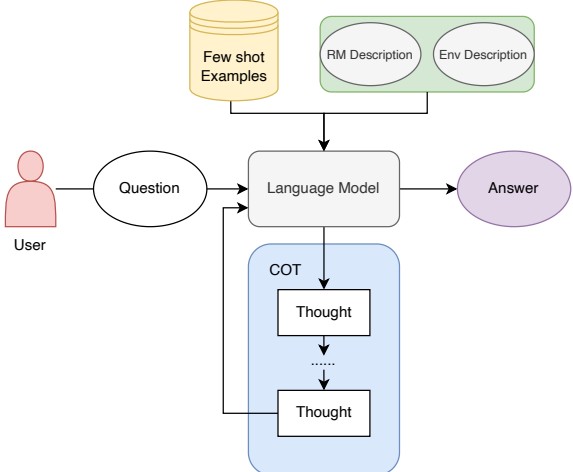

*Figure 8.* An overview of the framework to learn reward machine by large language model through domain knowledge.

# HalfCheetah

**Environment Description:** HalfCheetah is part of the Mujoco environments. This environment is based on the work by P. Wawrzyński in "A Cat-Like Robot Real-Time Learning to Run". The HalfCheetah is a 2-dimensional robot consisting of 9 body parts and 8 joints connecting them (including two paws). The goal is to apply a torque on the joints to make the cheetah run forward (right) as fast as possible, with a positive reward allocated based on the distance moved forward and a negative reward allocated for moving backward. The torso and head of the cheetah are fixed, and the torque can only be applied to the other 6 joints over the front and back thighs (connecting to the torso), shins (connecting to the thighs), and feet (connecting to the shins). The robot needs to go through 5 checkpoints c(x_position=2), e(x_position=4), f(x_position=6), g(x_position=8), and a(x_position=10). The agent gets a reward of 1000 when it reaches a and the control penalization (CP) otherwise. Agent can get its current location by calling self.info['x_position'].

**Question**: Design the set of propositional symbols $\mathcal{P}$ for HalfCheetah environment.

**Think**: To design the set of propositional symbols $\mathcal{P}$, first I need to find the key events in the environment. The robot need to go through 6 checkpoints c(x_position=2), e(x_position=4), f(x_position=6), g(x_position=8), and a(x_position=10). I can use them as the propositional symbols, if the agent reach some checkpoint, it will reach a new reward machine state. Besides, I need to consider situations when agent don't reach the checkpoints.

**Answer**: The set of propositional symbols can be a set including checkpoints in HalfCheetah-v3 and situations not reaching the next checkpoints . $\mathcal{P} = \{c, !c, e, !e, f, !f, g, !g, a, !a\}$

**Question**: Design the get event function for the HalfCheetah with reward machine.

**Think**: Here events is depended on the propositional symbols, so the events including reach the checkpoints, and call self.info['x_position'] can get the agent's current x_position. Compare it with checkpoints' x_position, we can get the events.

**Answer**:

```
def get_events(self):
    events = ''
    if self.info['x_position'] < -10:
        events += 'b'
    if self.info['x_position'] > 10:
        events += 'a'
    if self.info['x_position'] < -2:
        events += 'd'
    if self.info['x_position'] > 2:
```

```
        events += 'c'
    if self.info['x_position'] > 4:
        events += 'e'
    if self.info['x_position'] > 6:
        events += 'f'
    if self.info['x_position'] > 8:
        events += 'g'
    return events
```

**Question:** Design the reward machine for the HalfCheetah environment.

**Think:** To design the reward machine. I need to verify the reward machine states first. As $\mathcal{P} = \{c, !c, e, !e, f, !f, g, !g, a, !a\}$. The reward machine including 5 states. As the agent reach checkpoint a, the game is finished. So these is one terminal state. Thus the reward machine state is $\{0, 1, 2, 3, 4, 5\}$ based on $\mathcal{P}$. Besides, as the agent gets a reward of 1000 when it reaches a and the control penalization (CP) otherwise. The $\delta_r$ can be designed based on it.

**Answer:**

$$0 \quad \# \text{ initial state}$$
$$[5] \quad \# \text{ terminal state}$$
$$(0, 0,' !c', \text{RewardControl}())$$
$$(0, 1,' c', \text{RewardControl}())$$
$$(1, 1,' !e', \text{RewardControl}())$$
$$(1, 2,' e', \text{RewardControl}())$$
$$(2, 2,' !f', \text{RewardControl}())$$
$$(2, 3,' f', \text{RewardControl}())$$
$$(3, 3,' !g', \text{RewardControl}())$$
$$(3, 4,' g', \text{RewardControl}())$$
$$(4, 4,' !a', \text{RewardControl}())$$
$$(4, 5,' a', \text{ConstantRewardFunction}(1000))$$

## A.4. Additional Experiment

In order to present our experimental results more clearly and completely, we provide all the results of our experiments in this section of the appendix. In addition to the two sets of experiments given in the main text, we also provide a set of toy examples to show the migration between nchain tasks of different scales. In addition, we provide the results of the reward translation task under three different Mujoco tasks, including the Hopper-V3, HalfCheetah-V3 and Ant-V3.

### A.4.1. NCHAIN EXPERIMENT

In the NChain game, an agent navigates through a linear sequence of states (where N denotes the number of states), aiming to reach the flag's location and grab it. The agent receives a reward:

$$(r = 1 - \frac{\text{Used\_Steps}}{\text{Max\_Steps}}). \tag{11}$$

The agent will receive the reward only upon grabbing the flag. At each state, the agent has three possible actions:

$$\{\text{move forward}, \text{move backward}, \text{grab the flag}\}.$$

In this experiment, the NRT Framework attempts to transfer rewards from NChain(n=5) to NChain(n=9) (illustrated in Figure 9(a)). This experimental setup intends to verify the NRT model's capacity for accurately extracting abstract alignments from semi-alignable MDPs; despite differing in observation space size, NChian(n=5) and NChain(n=9) share an isomorphic reward machine based on correspondingly aligned checkpoints, which can be treated as propositional symbols. Figure 9(a)

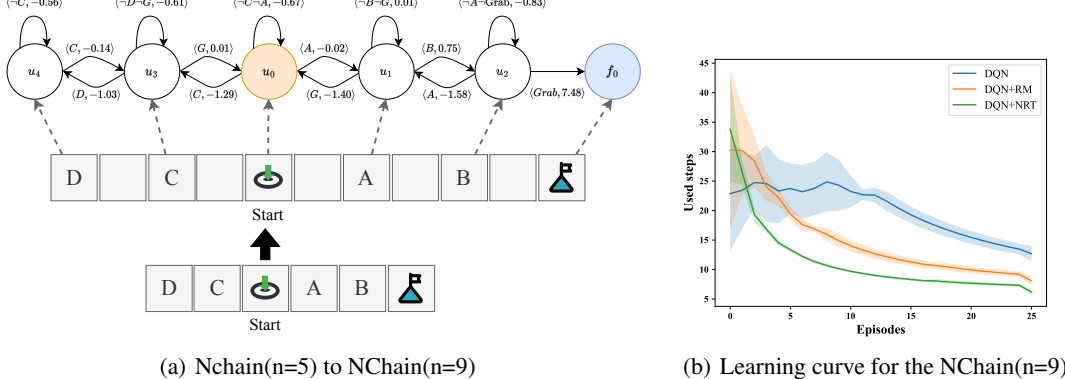

(a) Nchain(n=5) to NChain(n=9)  (b) Learning curve for the NChain(n=9)

*Figure 9.* Experiment for NChain game. (a) the environment setting and the isomorphic reward machine for NChain game, and the reward transition function is learned from NChain(n=5); (b) The learning curve for NChain game.

also demonstrates the shared isomorphic reward machine, with reward transitions learned from the source task NChain(n=5). This reward machine effectively guides the agent in completing the NChain(n=9) tasks, providing a clear preference for the correct path to reach the target. For this experiment, we utilized the DQN as a baseline, and the training results are illustrated in Figure 9(b). Our findings indicate that the NRT Framework can successfully extract abstract alignments from semi-alignable MDPs, while the transferred rewards from the isomorphic reward machine facilitate and promote training.

### A.4.2. 3D VISUAL NAVIGATION

In the 3D visual navigation environment, we selected the Sign task in the Miniworld (Chevalier-Boisvert et al., 2023) as the target task, with the Text-Sign task serving as the original task. Figures. 4 depict the original and target tasks, respectively. In the Miniworld-Sign environment, the agent receive a sparse reward only if it reaches all the targets including the red key, the blue box and the red box. The sparse reward is formulated as:

$$r = 1000 \times \frac{\text{Max\_Steps} - \text{Used\_Steps}}{\text{Max\_Steps}}. \tag{12}$$

At each step, the agent's observation is restricted to the information perceived in the direction of its movement, and it must

select from a limited action space consisting of four actions:

$$\{\text{turn left}, \text{turn right}, \text{move forward}, \text{move back}\}.$$

Due to the partial observability and sparse reward problem, solving Miniworld-Sign with conventional reinforcement learning methods proves challenging. By contrast, the Text-Sign task is simpler, as the agent can observe the entire map and directly choose actions to move up, down, left, or right, making task completion significantly easier. The goal of the experiment is to transfer knowledge from the trained agent in the Text-Sign environment to the Miniworld-Sign environment within isomorphic reward machines.

We also implemented a naive reward approach on the Sign task (shown in Figure 11, where each time the agent take one thing, it will get a reward +10), where the agent receives supplementary rewards when making progress toward the goal. The results show that while naive reward shaping does improve performance compared to sparse rewards, there remains a substantial performance gap when compared to our NRT method with transferred reward machines. This comparison highlights that while simple reward shaping can help, the structured knowledge transfer facilitated by NRT provides more substantial benefits for learning efficiency. The transferred reward structure itself inherently contains this shaping information but in a more principled way that captures the logical dependencies between subtasks, which explains the superior performance of our approach.

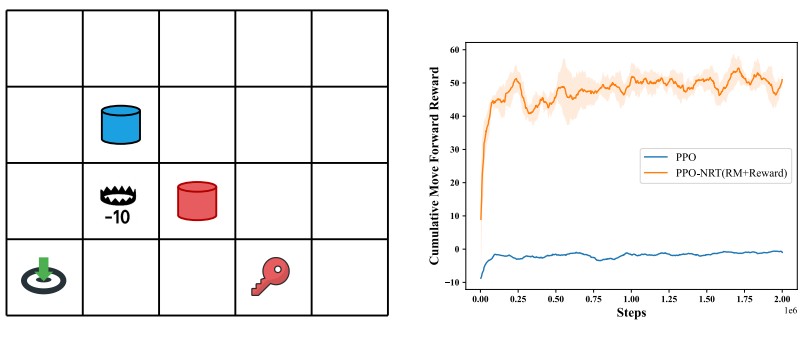

(a) Sign-with-Trap Environment Diagram     (b) Learning curve for Sign-With-Trap

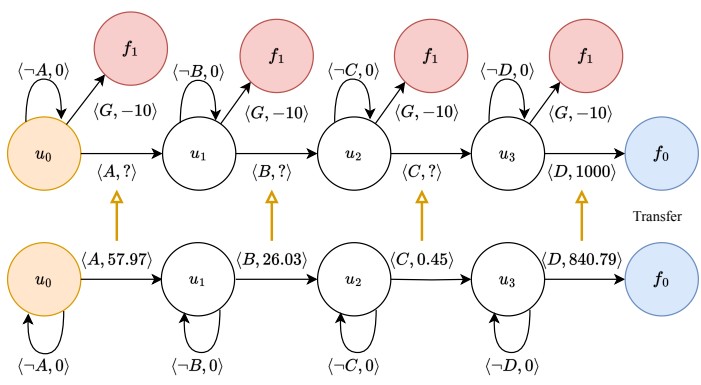

(c) Transferred reward machine from Text-Sign to Sign-with-Trap Environment

*Figure 10.* Experiment for Sign-with-Trap. (a) Sign-with-Trap Environment Diagram; (b) Transferred reward machine from Text-Sign to Sign-with-Trap Environment; (c) Learning curve for Sign-With-Trap

Furthermore, we have expanded our evaluation by modifying the Text-Sign task to incorporate more complex disjunctions. In this enhanced environment, we added a "trap" mechanism where the agent receives a -10 reward and terminates the episode if it falls into the trap at any point. From a reward machine perspective, this modification adds a branch from each

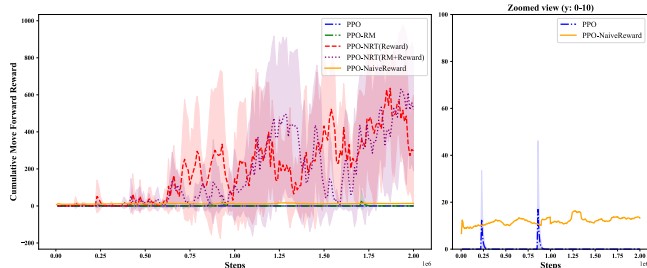

*Figure 11.* Learning curve for Miniworld-Sign-Add Naive Reward Basline

state (U0, U1, U2, U3) to a terminal failure state, creating multiple disjunctive paths through the task. The environment diagram and transferred reward machine is shown in Figure 10(a) and Figure 10(c) in the additional experiment. The results (shown in Figure 10(b)) demonstrate that NRT continues to show significant performance improvements when transferring the reward machine and corresponding rewards from the original Text-Sign task, even with these more complex logical structures.

### A.4.3. MUJOCO

In the Mujoco experiment, we selected HalfCheetah, Hopper, and Ant as the target tasks, with a simple NChain game serving as the source task. Figure. 12 depicts the environment used in this experiment. All target tasks employ the same settings as

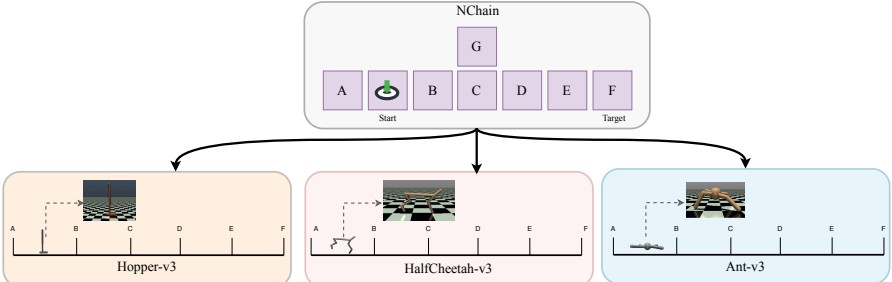

*Figure 12.* The environment for Mujoco environments. The source task is an NChain game where the agent need to reach the point F and get a reward. And Hopper-v3, HalfCheetah-v3 and Ant-v3 are the source tasks while their target point is F.

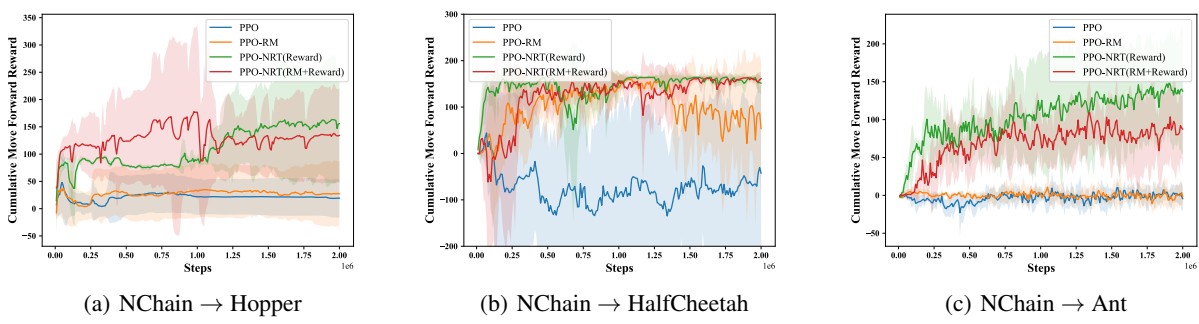

    (a) NChain → Hopper          (b) NChain → HalfCheetah         (c) NChain → Ant

*Figure 13.* The learning curves for mujoco experiment. To intuitively precept the learning process of the agent, we use the original reward provided by OpenAI-Gym which consists forward reward and control reward to show the learning curve.

their respective OpenAI-Gym versions (Brockman et al., 2016); however, we adjusted the rewards to be sparse, receiving $r = 1000$ only upon achieving specific goals. The agent in the source task navigates through a linear sequence of states to reach point F, earning a sparse reward equivalent to Equation. 12. At each state, the agent has four possible actions: move left, move right, move up, and move down. The generated reward machine and the cross-domain reward transfer process

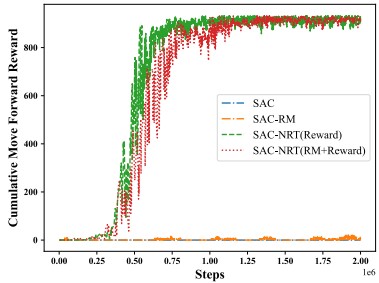

*Figure 14.* Learning curve for Mujoco-Ant on SAC method

from NChain to Mujoco are depicted in Figure. 7(b). The source task's reward machine and target tasks' reward machines are homomorphic reward machines. Our analysis suggests that the reward signal from the source task also provides guidance for the target tasks, similar to the NChain game experiment. Besides, compare the generated reward machine based on LLM and the hand-designed reward machine shown in Figure. 2(b), the reward machine generated by LLM maintains a same structure of the hand-designed reward machine, which shows the effectiveness of the reward machine generator based on LLM. The learning results are shown in Figure. 13. We still chose PPO as the baseline. PPO-RM, PPO-NRT(Reward), and PPO-NRT(RM+Reward) to perform an ablation study to exclude potential influences due to reward machine utilization. Our results demonstrate that in extremely sparse reward tasks, merely extending observation with reward machine information proves insufficient; in contrast, using transferred rewards significantly alleviates the sparse reward issue, enhancing learning performance. In Hopper-v3 and Ant-v3 tasks, PPO-NRT(Reward) achieves the best results, while in HalfCheetah-v3 tasks, PPO-NRT(Reward+RM) and PPO-NRT(Reward) attains similar performance.

To further show the versatility of NRT, we have expanded our experiments to include SAC (Soft Actor-Critic) as an additional baseline for the Mujoco tasks. The result is shown in Figure 14.

