# OpenReview forum: "Reward Translation via Reward Machine in Semi-Alignable MDPs"
_ICML.cc/2025/Conference — ICML 2025 poster_

### Official Review · Reviewer_18Bw · 2025-03-12

**Overall Recommendation:** 3

**Summary:**

This paper considers a setup where we want to transfer the reward from source domain to the target domain so that we can train RL agents in target domain where the reward design is tedious or difficult. But this will be difficult for many source-target domains because many domains don't share the same structure. So the paper introduces the idea of using the concept of semi-alignable MDP -- by designing a method that first generates reward machine and transferring the reward machine. Experiments are conducted on some simple 3D visual navigation tasks and OpenAI Gym tasks.

## update after rebuttal

My score is updated from 2 to 3 to reflect that authors provided additional experiments in the rebuttal. I recommended authors to provide further diverse experiments if the paper is accepted.

**Claims And Evidence:**

Experimental results are very weak for supporing the claims made in the paper. Actually results are quite noisy and not significant so that it's difficult to see the trend.

**Essential References Not Discussed:**

N/A

**Experimental Designs Or Analyses:**

Again, the setup is a bit contrived and the paper should consider more backbone RL algorithms. Using the source domain of NChain is interesting but the paper can be made more stronger by considering more realistic scenarios of considering more 'relevant yet different' domains other than NChain domain.

**Methods And Evaluation Criteria:**

For OpenAI Gym experiments, experiments are conducted on a bit of contrived setup where the reward is artificially made sparse.

One additional note is that PPO is not 'state-of-the-art' -- this is method from 2017! - and it is not clear why the proposed idea is evaluated only on PPO --  maybe having more experiments based on other backbone RL algorithms can make the results be more convincing.

**Other Comments Or Suggestions:**

The authors may want to check how they should write double quote in latex

**Other Strengths And Weaknesses:**

Other strengths
- Related work was very useful in understanding relevant works

Other weaknesses
- The paper is a bit weak in providing a motivation. It could be nice to provide a trivial example where it is easy to transfer rewards and it is hugely beneficial.

**Questions For Authors:**

- Would it be possible to provide experimental results on a more 'natural' setup where the reward is not artificially removed from the task? Removing the reward at all, thus making PPO baseline entirely fail, and then showing that the proposed idea can work may not be an ideal way of showing the promise of reward transition.
- It seems to me that PPO policies are all stuck in local minima for OpenAI Gym tasks -- is there a particular reason for this?
- Regarding the above point, is there any qualitative analysis that shows which reward is transferred?
- Would there be a way to generate reward machines without using LLMs?

**Relation To Broader Scientific Literature:**

Reward translation is an interesting idea that can be useful for many scenarios, but the current method is a bit too complex and does not have promising results yet.

**Theoretical Claims:**

I don't have an enough background / knowledge to thoroughly go through the proof.

---

> ### Author Rebuttal · Authors · 2025-04-01
>
> Thank you for your thoughtful review of our paper. We appreciate your feedback and the opportunity to address your concerns regarding our approach and experimental results. We have updated the additional experiment in:
> https://drive.google.com/file/d/1_U1d13bM4kG1reHUdx2wkLNb9zfn4otS/view?usp=sharing
> (Due to time constraints, many supplementary experiments could not be run with multiple groups to account for variance effects.）
> ## 1. Regarding baseline and experiment design.
> We appreciate this concern about our algorithmic choices. We must emphasize that our paper's primary contribution is not proposing a superior RL algorithm, but rather enabling cross-domain reward reuse - the RL method simply serves to validate our reward transfer effectiveness and should avoid excessive additional techniques. From this perspective, we selected PPO for several reasons:
> 1.	PPO is a widely-used, reliable algorithm that doesn't introduce additional techniques that might confound our results.
> 2.	Using a consistent algorithm across most environments allowed for standardized experimentation.
> 3.	PPO's compatibility with both discrete and continuous action spaces facilitated unified testing methodology.
> We should clarify that our baselines aren't limited to PPO alone - our Nchain2Nchain experiments utilized DQN. Additionally, to address your concern, we have expanded our experiments to include SAC (Soft Actor-Critic) as an additional baseline for the Mujoco tasks.The result is shown in Figure 4 in the additional experiment. These new results, included in our supplementary materials, demonstrate that our method's benefits are consistent across different RL algorithms, strengthening the generality of our approach.
> Our experimental design deliberately demonstrates knowledge transfer from simpler source tasks to more complex target domains, showing how fundamental task structures can inform learning in challenging environments. We carefully selected tasks to demonstrate effectiveness with both isomorphic and homomorphic reward machines, specifically addressing semi-alignable MDPs - a challenge inadequately addressed in previous research.
>
> Beyond MuJoCo, our Text-Sign to 3D-Sign transfer experiments validate our approach in first-person navigation with sparse rewards. To address concerns about reward structure, we ran additional experiments with naive reward shaping (Figure 5 in additional experiment). While simple shaping improves performance, our NRT method still outperforms by capturing logical dependencies between subtasks, providing more principled guidance.
>
> ## 2. Regarding reward visualization and transfer
> Our reward machine diagrams already indicate transferred reward values directly, showing how values map between domains - a more intuitive representation than alternatives like heat maps. We provide a visualization example of the reward machine heatmap from NChain to MuJoCo, as given in the link below, for your reference and comparison.
> https://drive.google.com/file/d/1wPJJbeZjK6_6ZTib4JEaBnueIoMja6nU/view?usp=sharing
>
> ## 3. Regarding performance in Mujoco environments
> You correctly noted that PPO policies seem to struggle in our modified Gym tasks. This is by design - we modified the standard Mujoco environments to use sparse rewards, where agents only receive feedback upon reaching specific checkpoints (e.g., point F when x-position exceeds 8), rather than the dense rewards in the original environments that provide immediate feedback at every step.
> We should emphasize that this checkpoint-based task setting for Mujoco environments has precedent in classical reward machine literature, specifically in Icarte et al.'s "Reward Machines: Exploiting Reward Function Structure in Reinforcement Learning." However, while Icarte et al. retained the control reward component from the original environments, we implemented a fully sparse reward setting to create a more challenging scenario that better highlights the benefits of our approach.
> This sparse reward setting intentionally creates a challenging learning scenario that better demonstrates the value of our approach. Standard PPO struggles because most sampled trajectories contain no reward signal. Our NRT method, while still operating in a relatively sparse reward regime, provides more informative guidance through the transferred reward structure, helping overcome this exploration challenge.
> ## 4. Regarding reward machine generation without LLMs
> While we used LLMs, alternatives include manual definition and Icarte et al.'s combinatorial optimization approach. We focused on the reward translation framework itself rather than reward machine generation, which could be explored in future work.
>
> Thank you again for your valuable feedback, which has helped us strengthen both our presentation and experimental validation.

---

> > ### Comment · Reviewer_18Bw · 2025-04-03
> >
> > Thank you for the response and I like that you added new experiments. I would strongly recommend adding similarly diverse experiments in the camera-ready if the paper is accepted; that would make the paper much stronger and interesting to the community.

---

> > > ### Author Response · Authors · 2025-04-07
> > >
> > > We sincerely thank you for your insightful and constructive comments that improved our work. We will add these diverse experiments.

---

### Official Review · Reviewer_i3La · 2025-03-13

**Overall Recommendation:** 3

**Summary:**

This paper proposes a way to derive reward functions for cross domain transfer learning. This is achieved via reward machines for obtaining a transferable reward in semi-align MDPs. Experiments conducted on 3D visual navigation and a few Mujoco tasks demonstrate the benefits when agents are learned with PPO.

**Claims And Evidence:**

The claims of this paper are around developing the foundations for reward translations and semi-alignable MDPs which are defined in the Section 3 and 4.

**Essential References Not Discussed:**

NA

**Experimental Designs Or Analyses:**

The experiments on Mujoco tasks are not convincing enough. The source domain is a NChain environment where the agent needs to reach a point F and is used to define a reward for target domain where the Ant / HalfCheetah needs to reach point F. This is a relatively simple task. What if the source task is more harder? Given that the paper uses LLMs (as described in Sec 4), will the method work when the source domain in HalfCheetah and the target domain is Ant.

**Methods And Evaluation Criteria:**

The paper evaluates on 3 tasks from Mujoco (HalfCheetah, Ant and Hopper) and on 3D visual navigation based on Mini`world.

**Other Comments Or Suggestions:**

NA

**Other Strengths And Weaknesses:**

NA

**Questions For Authors:**

1. The example in Fig 1 is not convincing. In the Half-Cheetah environment, the is run / walk with a certain speed and not to reach a point F as in NChain. How are these two tasks related and how is Point F defined in HalfCheetah environment?

2. The second condition (b-dynamic) in Def 3.1 is not clear to me. How does the function $\phi$ applies to $Pr^B$ as this is a transition function and should return probabilities?

3. In Definition 3.3, how the action domain x is used in defining the Reward Machine? It is mentioned, but not used anywhere. Why x is a action domain, in Sec 3.1 X is defined as a source domain?

4. What if instead of NChain, a cheetah task where the agent has to reach the point F is used for transfer such that the Ant reaches a point F?

**Relation To Broader Scientific Literature:**

The idea of recovering a reward function that can be transferred across domains is challenging. However, the experiments are not convincing enough.

**Theoretical Claims:**

The 2) condition in Definition 3.1 seems off. As the $\phi$ is applied over the transition function $Pr$. The theorem 3.5 looked fine. I did not check Theorem 4.5 closely.

---

> ### Author Rebuttal · Authors · 2025-04-01
>
> Thank you for your thoughtful review of our paper. We appreciate your careful reading and questions, which help us improve the clarity and rigor of our work. We have updated the additional experiment in: https://drive.google.com/file/d/1_U1d13bM4kG1reHUdx2wkLNb9zfn4otS/view?usp=sharing
> (Due to time constraints, many supplementary experiments could not be run with multiple groups to account for variance effects.）
> ## 1. Regarding the HalfCheetah example in Fig 1
> We apologize for any confusion in our presentation. To clarify, we have modified the standard Mujoco environments to create goal-oriented tasks that better demonstrate our cross-domain transfer approach. For the HalfCheetah environment, we defined checkpoints along the x-axis, with point F specifically representing when the robot's position exceeds x=8. This task definition follows similar approaches in Icarte et al.'s work Reward Machines: Exploiting Reward Function Structure in Reinforcement Learning (which we have cited).
> The key difference from the standard HalfCheetah environment is that we use a sparse reward structure where the agent receives a reward only upon task completion (reaching point F), with the reward magnitude depending on the number of steps taken. This contrasts with the original dense reward setting that provides control and forward rewards at each step. This sparse reward formulation makes the task more challenging to learn but better demonstrates the value of our approach.
> ## 2. Regarding condition (b-dynamic) in Definition 3.1
> Thank you for requesting clarification on this point. The essence of this definition relates to our abstraction of environments into subgoals and skills, creating a coarser granularity than the standard RL state-action level.
> To elaborate: in environments like our Text-Sign task, the agent must collect specific items in sequence to complete the task. Each item collection represents a subgoal, and the process of achieving that subgoal constitutes a skill. This abstraction doesn't consider the specific actions or states involved in executing the skill (which would be fine-grained), focusing instead on the higher-level completion of subgoals.
> The mappings φ and η represent these coarse-grained alignments of subgoals and skills between source domain X and target domain Y. Our "semi-alignment" concept acknowledges that fine-grained alignment between domains is often infeasible, but partial reward reuse through these higher-level mappings remains valuable. In our implementation, this is achieved through reward machine states and propositional symbols mappings. The function Pr in Definition 3.1 is a deterministic function that indicates which new abstract state (equivalently, which reward machine state) the agent will transition to when selecting a particular skill from its current abstract state.
> ## 3. Regarding Definition 3.3 and domain notation
> We sincerely apologize for this notational error. You are correct that in Definition 3.3, "x" should be "A" (representing the action space), not the source domain. This is indeed a typographical error that we will correct in the final version.
> ## 4. Regarding experimental concerns
> The core contribution of our work is enabling reward translation for cross-domain transfer, with reward machines as the bridging mechanism. Our experiments focus on transferring knowledge from simpler tasks (e.g., NChain) to more complex target domains, demonstrating how fundamental task structures can inform learning in challenging environments.
>
> We carefully selected tasks to highlight our approach’s effectiveness in both isomorphic and homomorphic reward machine settings, specifically addressing semi-alignable MDPs—an area insufficiently explored in prior work. Beyond Mujoco, we also validated our method in a significantly different domain: transferring from Text-Sign to 3D-Sign, where first-person navigation with sparse rewards poses a unique challenge.
>
> Regarding Mujoco transfers (e.g., HalfCheetah to Ant), our method is applicable but unnecessary from a motivation standpoint. These environments share the same reward machines, and existing work (e.g., Raychaudhuri et al. Cross-domain Imitation from Observations) has already demonstrated effective cross-domain transfer for alignable MDPs (They also transfer HalfCheetah to Ant). Since our approach abstracts reward structures rather than leveraging shared state-action similarities, it does not offer additional benefits in such cases. That said, due to their inherent similarity, even with our method, performance would be better than in NChain.
>
> We believe these clarifications address your concerns and hope they provide a clearer understanding of our approach and its contributions. We thank you for your valuable feedback, which will help us improve the presentation of our work.

---

### Official Review · Reviewer_H5bA · 2025-03-18

**Overall Recommendation:** 4

**Summary:**

The paper introduces the Neural Reward Translation (NRT) framework, a novel methodology designed to transfer knowledge from completing a task in one environment to quickly learning to solve a (sufficiently similar) task in another environment. For example, NRT can transfer knowledge gained from completing a task in a grid with discrete actions to learning how to solve a similar task in a 3D environment with continuous actions. To achieve this, NRT utilizes the optimal value function of the original task (e.g., the grid world) to shape the reward for the target task (e.g., the 3D environment). Furthermore, the paper formally defines the concept of semi-alignable MDPs. Based on this definition, the authors demonstrate the conditions under which NRT can be used to transfer knowledge from one MDP to another.

**Claims And Evidence:**

Yes

**Essential References Not Discussed:**

I think all the important works are being discussed.

**Experimental Designs Or Analyses:**

Yes, the experimental design and analysis are sound.

**Methods And Evaluation Criteria:**

Overall, I believe the experimental evaluation is solid. However, I have two concerns regarding the experiments.

1) I find it somewhat disappointing that NRT was tested only on sequential tasks. It would have been great to see tasks that include disjunctions, conjunctions, or cycles (beyond self-loops). That said, I don’t believe there’s any reason the proposed method wouldn’t perform well in those cases as well.

2) I find it challenging to assess the merits of the proposed method without additional baselines. For example, we could shape the reward function by providing extra rewards whenever the agent makes progress in the task (i.e., moves closer to the terminal state in the RM). I believe that such a baseline could be competitive with the proposed method, and it is straightforward to implement.

**Other Comments Or Suggestions:**

I suggest better explaining $\Pr^B$ in Section 3.2. Currently, it just states that it “_denotes the transition on goal._” However, if I understand correctly, it represents a probability from $B \times W$ to $\Pr(B)$. If that is the case, please make that explicit. Also, I find it strange that in Definition 3.1 the paper states that $\Pr^B_y(b_y,w_y) = \phi(\Pr^B_x(b_x,w_x))$ because that implies that $\Pr^B_x(b_x,w_x)$ is actually a deterministic function (not a probability distribution) that returns one state $b \in B_x$ (since $\phi$ was defined as a function from $B_x$ to $B_y$).

In Sections 3.3 and 4, I am confused by the meanings of $P$, $\mathcal{P}$ and $2^{\mathcal{P}}$. Usually, $\mathcal{P}$ refers to the set of propositional symbols and $2^{\mathcal{P}}$ is a truth value assignment to each of those symbols. So, for instance, in one transition, symbols $a$ and $b$ could hold at the same time. Thus, the definition of $\delta_u$ in an RM goes from $U \times 2^{\mathcal{P}} \mapsto U$. But the paper defines them in terms of $U \times P \mapsto U$. So, I don’t know if that is a typo, or if $P$ is used as shorthand notation for $2^{\mathcal{P}}$. Then in Theorem 3.5, I think $\Gamma_x$ should go from $2^{\mathcal{P}_x}$ to $W_x$ because in every RM transition, multiple propositions might hold at once.

**Other Strengths And Weaknesses:**

Overall, I think this is a solid paper. It explores an important issue: how to transfer knowledge from one domain to another. To do so, it formally defines semi-alignable MDPs and then uses RMs to develop a practical algorithm. The results, although limited to sequential tasks, show that this novel method can work well in practice.

I only have two minor concerns about this work:

1. The paper discusses the potential use of LLMs to automatically generate an RM. However, I wouldn’t consider this a major contribution of the work because the method itself is not well explained, and it is unclear whether it would be effective in a different domain.

2. I suspect that a simple reward shaping technique could perform just as well as the proposed method for the sequential tasks tested in the paper. As such, the experimental section would be more compelling if it assessed other types of tasks, particularly those involving cycles or disjunctions. Additionally, I believe that incorporating a naive reward shaping method as a baseline would enhance the paper. Even a straightforward approach, such as giving a reward of +100 every time the agent makes progress on the task, would help better evaluate the effectiveness of the proposed method.

**Questions For Authors:**

1. If we use a naive reward shaping technique, such as always providing an extra reward for making progress on the task—or something similar to the automated reward shaping method proposed by Icarte et al. (2022)—would that work similarly to the proposed method? Or would NRT work better? Why?

2. In what way is the proposed method superior to the automated reward shaping suggested by Icarte et al. (2022)?

**Relation To Broader Scientific Literature:**

The paper explains that previous attempts to transfer rewards would not be effective with incompatible MDPs (which are non-pairable and non-time-alignable). However, it does not connect to the literature on automated reward shaping or the literature on intrinsic motivation. I believe that any method altering the reward function could serve as a valid alternative for using NRT.

**Theoretical Claims:**

The definitions and theorems appear correct to me. However, I have a couple of comments, which are discussed below.

---

> ### Author Rebuttal · Authors · 2025-04-01
>
> We deeply appreciate your thoughtful review and constructive feedback on our NRT paper. We have updated the additional experiment in:
> https://drive.google.com/file/d/1_U1d13bM4kG1reHUdx2wkLNb9zfn4otS/view?usp=sharing
> (Due to time constraints, many supplementary experiments could not be run with multiple groups to account for variance effects.）
> Your suggestions will certainly strengthen our work, and we address each point below:
> ## 1. Regarding testing NRT beyond sequential tasks
> Thank you for this valuable suggestion. While the NChain to Mujoco task does contain a basic disjunction (at point C, the agent can choose to go to point G or point D), we acknowledge this implementation is relatively simple.
> We have now expanded our evaluation by modifying the Text-Sign task to incorporate more complex disjunctions. In this enhanced environment, we added a "trap" mechanism where the agent receives a -10 reward and terminates the episode if it falls into the trap at any point. From a reward machine perspective, this modification adds a branch from each state (U0, U1, U2, U3) to a terminal failure state, creating multiple disjunctive paths through the task. The environment diagram and transferred reward machine is shown in Figure 1 and Figure 2 in the additional experiment.
> Our results (shown in Figure 3 in the additional experiment) demonstrate that NRT continues to show significant performance improvements when transferring the reward machine and corresponding rewards from the original Text-Sign task, even with these more complex logical structures.
> ## 2. Additional baselines (naive reward)
> We agree that additional baselines strengthen our evaluation. We implemented a naive reward approach on the Sign task (shown in Figure 5 in the additional experiment, where each time the agent take one thing, it will get a reward +10), where the agent receives supplementary rewards when making progress toward the goal. The results (available in our supplementary materials) show that while naive reward shaping does improve performance compared to sparse rewards, there remains a substantial performance gap when compared to our NRT method with transferred reward machines.
> This comparison highlights that while simple reward shaping can help, the structured knowledge transfer facilitated by NRT provides more substantial benefits for learning efficiency. The transferred reward structure itself inherently contains this shaping information but in a more principled way that captures the logical dependencies between subtasks, which explains the superior performance of our approach.
> ## 3. On automated reward machine generation
> Thank you for highlighting this concern. We want to clarify that our primary contribution is the reward translation framework (cross-domain reward transfer), with reward machines serving as the bridge for this translation process.
> Icarte et al.'s work on automated reward machines is indeed excellent, formulating a combinatorial optimization problem for reward machine construction. However, solving this requires substantial data and computational resources. Our LLM-based approach to generating reward machines is admittedly more similar to human specification but offers practical automation benefits.
> We agree that combining Icarte's automated reward machine learning with our reward translation framework represents a promising direction for end-to-end learning. This integration would need to address challenges like limited trajectory data in target tasks and modeling complexities in continuous action spaces like Mujoco. But it is far from what this paper aims to discuss.
> ## 4. Clarification on propositional symbols (P) and definition 3.1
> We apologize for any confusion regarding our notation. P represents the set of propositional symbols, expressed as textual predicates (e.g., "c" or "!c" indicating whether the agent has or hasn't reached point c). The notation 2^P refers to the power set of these symbols, representing all possible truth assignments. We should have used 2^|P| for clarity, and we will correct this in the camera-ready version if accepted. The function Pr in Definition 3.1 is a deterministic function that indicates which new abstract state (equivalently, which reward machine state) the agent will transition to when selecting a particular skill from its current abstract state.
>
> We sincerely thank you for your thoughtful review and hope our responses address your concerns.

---

> > ### Comment · Reviewer_H5bA · 2025-04-01
> >
> > Thank you for your response and the additional experiments. My primary concerns have been addressed, so I have increased my recommendation score to Accept.

---

> > > ### Author Response · Authors · 2025-04-07
> > >
> > > We sincerely thank you for your insightful and constructive comments that improved our work.

---

### Decision · Program_Chairs · 2025-05-01

**Decision:**

Accept (poster)

**Comment:**

The submission focuses on an important problem of how to transfer knowledge from one domain to another. The authors focus on this by formally defining semi-alignable MDPs and then uses reward machines to develop a practical algorithm. The results, although limited to sequential tasks, show that this novel method can work well in practice.

There were reviewer concerns about a lack of comparison to other reward shaping baselines and the environments being too simplistic. However, it does appear as though the environments were designed to test the assumptions posed in the work, and I believe the contribution of transfer in a theoretically motivated semi-alignable MDP setting is significant.